# Relation of Dietary Patterns and Nutritional Profile to Hepatic Fibrosis in a Sample of Lebanese Non-Alcoholic Fatty Liver Disease Patients

**DOI:** 10.3390/nu14122554

**Published:** 2022-06-20

**Authors:** Nicole Fakhoury Sayegh, Gessica N. H. A. Heraoui, Hassan Younes, Lea Nicole Sayegh, Christa Boulos, Raymond Sayegh

**Affiliations:** 1Department of Nutrition, Faculty of Pharmacy, Saint Joseph University, Damascus Road, Riad el Solh, Beirut P.O. Box 11-5076, Lebanon; h.gessica10@gmail.com (G.N.H.A.H.); christa.boulos@gmail.com (C.B.); 2College Health, équipe PANASH-ULR 7519, Institut Polytechnique UniLaSalle, 19, Rue Pierre Waguet, CEDEX, 60026 Beauvais, France; hassan.younes@unilasalle.fr; 3Faculty of Medicine, American University of Beirut, Beirut P.O. Box 11-0236, Lebanon; lea.n.sayegh@gmail.com; 4Department of Gastroenterology and Hepatology, Faculty of Medicine, Saint Joseph University, Damascus Road, Riad el Solh, Beirut P.O. Box 11-5076, Lebanon; drrsayegh@gmail.com

**Keywords:** hepatic fibrosis, dietary patterns, the traditional diet, the high fruit diet

## Abstract

Non-alcoholic fatty liver disease (NAFLD) is considered the most common liver injury worldwide. NAFLD can evolve into non-alcoholic steatohepatitis (NASH) with or without fibrosis. The objectives of this study were to determine the nutritional profile and dietary patterns of NAFLD Lebanese patients and to report the type of diet-related to the presence of hepatic fibrosis. We hypothesized that the traditional pattern was related to a low risk of fibrosis. This cross-sectional study included 320 eligible Lebanese NAFLD patients. Three dietary patterns were identified: the Traditional diet, the High Fruit diet, and the Westernized diet. Multivariate analysis showed a significant relationship between high adherence to the traditional diet and absence of hepatic fibrosis with a decreased risk of 82%, *p* = 0.031 after adjusting for its covariables. Fruits were absent from this dietary pattern. Although our results pointed to a possible relationship between fibrosis in NAFLD patients and fruit intake, experimental studies are needed to show whether this is a causal relationship. However, the results obtained in this study may contribute to the planning of dietary interventions and recommendations and enable a better follow-up for NAFLD patients with fibrosis.

## 1. Introduction

Non-alcoholic fatty liver disease (NAFLD) is considered the most known liver injury in the USA and probably worldwide [1]. The simplest form of this disease is the pure fatty liver, which is characterized by the accumulation of triglycerides in hepatocytes at a percentage greater than 5% of the total number of cells [2]. The prevalence of the disease varies remarkably according to the diagnostic tools (liver biopsy, radiological tests, and blood tests) and the chosen population [2]. It is approximately 20 to 30% in the Middle East and in obese patients, the prevalence has ranged from 50 to 90% [3]. Concerning gender, some studies have shown that the prevalence of the disease is two to three times higher in men than in women [4].

NAFLD can progress into non-alcoholic steatohepatitis (NASH), which is defined as steatosis associated with inflammatory lesions and hepatocyte ballooning with or without fibrosis [5]. NASH can progress into cirrhosis and ultimately into hepatic cellular carcinoma [5]. Recently, studies have shown a positive association between the characteristics of the metabolic syndrome and NAFLD/NASH, especially abdominal obesity, hyperglycemia, hypertriglyceridemia, dyslipidemia, and arterial hypertension [6]. Non-alcoholic fatty liver disease is recognized as the liver manifestation of metabolic syndrome [6].

A biopsy is considered the most reliable method for detecting the presence of NAFLD and for assessing the inflammatory state of the liver as well as the presence or absence of fibrosis [7]. Nevertheless, it is rarely used in clinical practice because of its invasive nature and potential complications. Other non-invasive methods have emerged, such as liver biological tests and especially the NAFLD Fibrosis Score (NFS), which is defined as follows:$$\;\mathrm{NFS}\;\mathrm{score}\;=-1.675+0.037\times age\;\left(years\right)+0.094\times BMI\;(\mathrm{kg}/m^{2})+1.13\times diabetes\;\left(yes/no\right)+\;0.99\times ASAT/ALAT-0.013\times platelet\;count\;\left(109\;\mathrm{Giga}/L\right)\times \left(-0.66\right)\;albumin\;\left(g/\mathrm{dL}\right).$$

Values less than −1.45 indicate a state of non-fibrosis, values between −1.45 and 0.67 indicate a possible presence of fibrosis, and values above 0.67 indicate advanced fibrosis [8].

Dietary patterns have a significant impact on the biological and physical profile of patients [5]. The use of dietary patterns in establishing an association between dietary intake and NAFLD, mainly in its advanced stage, is of importance. This will enable the investigation and measure of the overall diet, which is more likely to be associated with NAFLD than each studied nutrient or single food [9].

Currently, there is no data on the prevalence of NAFLD in Lebanon and no studies describe the nutritional profile of NAFLD Lebanese patients with fibrosis. This observational cross-sectional study could be a preliminary study for more powerful ones such as cohorts and experimental studies. The main objectives were to analyze the dietary patterns and nutritional profile of Lebanese NAFLD patients in relation to the absence or presence of hepatic fibrosis. The main hypothesis was that the traditional diet was inversely associated with the presence of fibrosis.

## 2. Materials and Methods

### 2.1. Study Design

From November 2014 to June 2019, 500 Lebanese participants aged between 18 and 70 years, visiting an outpatient clinic of the department of gastroenterology in an academic hospital in Beirut, were invited to participate in the study after providing their informed consent. Four hundred patients diagnosed with NAFLD were recruited according to the inclusion criteria. Three hundred and twenty Lebanese patients completed the clinical, anthropometric, and dietary data. The response rate was 80%, which is statistically satisfactory. Figure 1 summarizes the course of the study. This sample size corresponded to the expected proportion (30%) of NAFLD patients with a confidence interval of 95% and an inaccuracy gap of 5% [10].

### 2.2. Sample Selection, Inclusion Criteria

Eligible patients were Lebanese men and women without: (1) biliary diseases or recognized cirrhosis; (2) infection with hepatitis A, B, or C virus; (3) genetic metabolic disease; (4) auto-immune liver diseases; (5) type 1 diabetes. Other inclusion criteria were (6) non-pregnancy among women; (7) less than or equal to two servings/day of alcohol consumption; and (8) the absence of drugs inducing hepatotoxicity (tamoxifen, steroids, amiodarone).

### 2.3. Study Protocol

The 320 patients were recruited and subsequently solicited for a face-to-face interview. These patients were recognized as having NAFLD following abdominal ultrasound by the same radiologist and with the same equipment (Hitachi-Aloka ProSound F75, Tokyo, Japan). The liver steatosis was estimated with the evaluation of the image brightness of the echo pattern. Abdominal ultrasound cannot identify hepatic fat deposition if it is less than 33% of the total liver and, accordingly, all patients with a lower percentage were categorized as free of the disease. A blood test, after a 10-h overnight fast, was requested during the interview with the patients after obtaining their informed consent. Serum samples were obtained from the coagulated blood after centrifugation and were immediately stored at −20 °C. Serum fasting blood glucose (mmol/L), triglycerides (mmol/L), ALAT (U/L), ASAT (U/L) HDL-C (mmol/L) and Albumin (g/dL) were assayed spectrophotometrically on Cobas c 501 (Roche, Germany). The serum insulin level (U/L) was measured by Electrochemiluminescence immunoassay (ECLIA) on a Cobas c 501 (Roche, Germany) and serum platelet count (Giga/L) was determined by Yumizen H2500/H1500 (Horiba, Japan). The reference values of the different biological parameters such as transaminases, albumin, platelets, and insulin corresponded to the reference values of the hospital laboratory. The blood pressure (mmHg) was taken using a manual blood pressure monitor (Bokang, China, 2009).

### 2.4. Measuring Tools

Other measuring tools included a face-to-face interview and anthropometric measurements. Dietary information was obtained using two questionnaires: a food frequency questionnaire (Harvard, Nurses’ Health Study, 2016) [11] translated into Arabic and adapted to Lebanese food, and two 24-hr recalls which summarize typical weekdays and weekend days (Appendix A) It was administered by the same dietician throughout the study, thereby limiting inter-investigator errors. It included the quantitative variables such as age (years), body mass index (BMI) (kg/m^2^) and socio-demographic qualitative variables such as gender, marital status, place of residence and birth, and occupation. Other qualitative questions referred to family history of obesity (limited to first-degree relatives), the presence of diabetes mellitus and dyslipidemia, and drug and supplement intake during the six months prior to diagnosis.

Questions about practicing physical activity, smoking status, and the number of cigarettes/day (qualitative variables) were asked. Physical activities were classified into two categories: mild/moderate or vigorous, according to the Centre for Disease Control Guidelines (CDC, 1993) [13]. Activities such as such as walking, housework, and gardening were considered mild or moderate (3.5 Kcal/min), while tennis, football, or fast swimming were considered vigorous (7 Kcal/min). Patients were vigorously active if the activity was vigorous and exceeded 20 min/day and was performed at least three times/week. It was considered moderate if the activity was moderate and exceeded 30 min/day and was performed at least five times/week and was during the last three months before the start of the study [14].

Photos with portion measurements (Numed, s. a. r. l) were used to determine the daily portion of different foods [15]. Certified software (Nutrilog, France, version 2.33) was used to analyze and determine the composition of different foods in micro and macronutrients. The frequency category selected for food (monthly, weekly, or daily) has been converted into a daily intake. The software analyzed the total daily energy intake of patients and the dietary composition of micro and macro-elements of different foods consumed. Fructose, ω-3, ω-6, Eicosapentaenoic acid (EPA) and Docosahexaenoic acid (DHA) intake was calculated according to the US Department of Agriculture (USDA) [16]. The database (1970) of the American University of Beirut was used for some national ingredients or recipes. Food fructose was analyzed as free fructose and/or as fructose from sucrose. Simple sugar has been defined as disaccharides and/or monosaccharides naturally present in foods or added sugar in commercial foods.

Questions such as the type and frequency of dietary supplements and vitamins, the frequency of fruit portions, soft drinks, and others (weekly, monthly, or daily), and the type of oil used in cooking were asked. To minimize response biases, the questionnaire included duplicate questions and patients were interviewed prior to their knowledge of the presence of fatty liver (Appendix A).

Anthropometric measures such as weight (kg) and height (cm), BMI (kg/m^2^) were taken by the same nutritionist and the same mechanical balance and stadiometer. The waist circumference (cm) and the hip circumference (cm) were taken by the same calibrated band throughout the study (average of three consecutive measures for each variable). Patients were classified as obese, overweight, or normal weight according to the WHO obesity classification, 2004: normal weight (BMI between 18 and 24.9 kg/m^2^), overweight (BMI between 25 and 29.9 kg)/m^2^) and obese (BMI > 30 kg/m^2^) [17].

Insulin resistance was studied by an evaluation index, the Homeostasis Model Assessment of Insulin Resistance (HOMA-IR). This index is calculated using the following formula: HOMA-IR = fasting plasma glucose (mmol/L)×fasting serum insulin (U/l)/22.5. A value greater than 3 indicates a state of insulin resistance [18]. Parameters that met the criteria of the International Diabetes Federation (IDF 2009) were identified to categorize patients with or without metabolic syndrome [12]. All clinical parameters were analyzed at the hospital laboratory using standard methods. Patients who were on drugs for hypertension, hyperglycemia or dyslipidemia were classified with metabolic syndrome regardless of laboratory findings.

### 2.5. Validity and Reproducibility of the Food Frequency Questionnaire

The reliability and validity of the Food Frequency Questionnaire (FFQ) were tested in a previous study [19]. The FFQ questionnaire was adapted from the Harvard Nurses’ Health Study, 2016, taking the general format (once/month, once/week, two to three times/week etc.). It was translated into Arabic and modified to Lebanese food items. It was administered into the native language, Arabic, and in a face-to-face method, by the same nutritionist throughout the study. The reproducibility of the FFQ was confirmed by administering it to 50 patients at the start of the study, prior to diagnosis. The re-interview of this subsample by the same nutritionist after one month yielded an interclass correlation co-efficient (ICC) = 0.957 (0.917–0.978), *p* = 0.0001 for energy intake/day of all participants. According to macronutrient intake/day, such as the percentage of carbohydrates and proteins of total energy intake/day, ICC varied between 0.969 (0.939–0.984) and 0.961 (0.924–0.980), respectively (*p* = 0.0001).

This coefficient corresponds to the agreement in energy intake/day (kcal/day) and macronutrients intake/day (g) or in their percentage of the total energy intake at two time points for each participant.

The estimate of validity was performed, using Bland-Altman analyses, on 100 patients, prior to diagnosis, who fulfilled both the FFQ questionnaire and the two 24-h recalls. The difference in dietary intake between the FFQ and the mean of estimated nutrients of both 24-h recalls was plotted on the Y axis, and the mean intake of both tools on the X axis. Most data points were clustered around the mean difference line between the two limits of agreement [19].

### 2.6. Ethical Considerations

The present study has complied with all ethical principles according to the revised Helsinki Declaration, 1975. The study has guaranteed the confidentiality and anonymity of the data. An informed consent letter was distributed and signed by the patients. The study started after obtaining the consent of the Ethics Committee of Saint Joseph University of Beirut (CEHDF 351) and was supported with grants by the research council of Saint Joseph University, Lebanon (FPH 34).

### 2.7. Analysis Plan and Statistical Tests

Descriptive data such as age, BMI, and waist circumference were presented as means and standard deviation. Categorical variables, such as gender, family history of obesity, cardiovascular disease (CVD), or type 2 diabetes were presented as frequency and percentage. Correlational studies were done by the chi-square test (bivariate study). Log10 of all quantitative variables with non-normal distribution was used. Logistic regression identified the main independent predictors of the absence or presence of fibrosis after adjustment for covariates. The score of each dietary pattern was entered as an independent variable with other co-variables. The threshold of significance was set at 5%. A statistical analysis was performed on SPSS 20 for Windows (IBM Corp., Released 2011, IBM SPSS Statistics for Windows, version 20.0. Armonk, NY, USA, IBM corp.)

Factor analysis was applied to extract dietary patterns from the FFQ. Food items were grouped into 25 groups according to food family and nutrient profile (Appendix A). The total consumption for each food group was set by determining the daily intake of servings from each item in this group. by The Kaiser Meyer-Olkin measure of sampling adequacy value was 0.788. The Bartlett’s test of sphericity value was significant (*p* < 0.0001). The number of components to extract was based on the Kaiser criterion (eigenvalues > 1), the change in the shape of the scree plot, and the loading of the items in the components generated (component matrix). Varimax rotation was conducted, and dietary patterns were named according to food groups with a factor loading greater than 0.2. Each participant had a factor score for each dietary pattern. The factor score for each participant was calculated by the Anderson and Rubin (1956) method [20]. The Traditional Lebanese pattern represents a group of related variables such as vegetables, chickpeas, red beans, olive oil, almonds, walnuts, and fish with a correlation matrix of r > 0.3. The High Fruit Pattern represents a group of related variables, all belonging to fruit families, such as plums, peaches, apricots, and apples. The variables of The Westernized pattern which represent an r > 0.3 were mainly beef meat, chicken, chips, carbonated beverages, pizza, and hot dogs. We also categorized patients into low, medium, and high adherence to each nutrient and dietary pattern based on tertiles of nutrients and dietary pattern scores. The association of dietary patterns and nutrients with hepatic fibrosis status (absence/presence or at risk) was determined using binary logistic regression. We defined three models with the crude one to compute multivariable–adjusted odds ratios (ORs).

## 3. Results

### 3.1. Variation of Age, Sociodemographic, Clinical, and Biochemical Parameters of the NAFLD Patients between the NFS Group (<−1.45/≥−1.45)

Values of the NFS score which are less than −1.45 indicate a state of non-fibrosis, while values ≥ −1.45 indicate a possible fibrosis [8]. The mean age (years) of the sample was 43.34 ± 0.12 (Table 1), and the risk of fibrosis was associated with increasing age (*p* = 0.0001). However, no significance was found in the risk of fibrosis according to sex, environmental, social status, and education. In total, 34.7% of participants lived in the Mount Lebanon district and 24.1% lived in Beirut. The remaining participants (35.9%) lived in the various Lebanese regions such as the North, South, Nabatiyeh, and Beqaa, with 5.3% living abroad. Regarding social status and education, 75.9% of the participants were married, 52.8% were university students and 0.9% were illiterate. The risk of fibrosis was significant according to the profession; 43.1% were self-employed, 31.2% were employees, and 21.2% were retired or unemployed (*p* = 0.001). There was no association between high economic status and risk of fibrosis, and 77.1% of participants had a crowding index (amount of residents/number of rooms) ≤1 (Table 1).

In total, 63.7% of the sample were obese (Table 1) and 83.9% had metabolic syndrome parameters. In addition, 21.2% had type 2 diabetes and 49.4% had Homa-IR > 3. As for family history, 84.7% of patients had a family history of cardiovascular diseases and/or diabetes, and/or obesity (Table 1). The presence of obesity, CVD, diabetes type 2 as well as the ratio of ASAT/ALAT were significantly different according to the NFS scores (*p* < 0.05).

### 3.2. Dietary Patterns

In this study, three dietary patterns were generated: The Traditional Lebanese, the High Fruit, and the Westernized diet. In total, 33.70% of NAFLD patients followed the Traditional diet (40.60% females, 30.20% males), (Figure 2). The adherence was almost the same for the High Fruits diet group (total 33.70%; 33.90% males, 33.30% females) and the Westernized diet group (total 32.60%; 35.90% males, 26.10% females). The Traditional diet contributed to 18.22% of the total variance. The High Fruit diet accounted for 9.84% of the total variance and the Westernized diet explained 7.22% of the total variance (Table 2).

### 3.3. Association between Dietary Patterns and Sociodemographic, Environmental, and Clinical Characteristics in the Study Population

The traditional pattern was associated with aging (years) (OR: 1.60; 95% CI: 1.03–1.09) which was inversely associated with a westernized one (OR: 0.92; 95% CI: 0.89–0.95), (Table 3). The latter was positively associated with an increase in BMI, *p* < 0.05. The traditional pattern was highly associated with the university level (OR: 2.90; 95% CI: 1.50–5.76), which was inversely associated with a High Fruit diet (*p* < 0.05).

### 3.4. Hepatic Fibrosis across Tertiles (T) of Dietary Pattern Scores

The risk of hepatic fibrosis across tertiles of the three dietary pattern scores is presented in Table 4. The association between the high adherence versus low adherence to the traditional pattern became statistically significant after adjustment for confounding variables. In the crude model, high adherence versus low adherence to the traditional pattern was associated with low odds of fibrosis (OR: 0.36; 95% CI: 0.13–1.03). In the third model, after further adjustment with the confounding variables, a significant decrease in the trend of odds of fibrosis was reported with high adherence versus low adherence to the traditional pattern (OR:0.18; 95% CI: 0.04–0.85). The other dietary patterns (High Fruits and Westernized patterns) showed no significant association with hepatic fibrosis, *p* > 0.05.

### 3.5. Hepatic Fibrosis across Tertiles of Food Groups

A significantly lower risk of fibrosis was observed with medium intake of EPA and DHA (tertile 2) versus low intake respectively (OR: 0.35; 95% CI: 0.15–0.82), (OR: 0.26; 95% CI: 0.09–0.73). A significantly higher risk of fibrosis was observed with medium intake of ω-6 (g) and simple carbohydrates (g) as compared to the low intake of these two nutrients (OR: 2.24; 95% CI: 1.06–4.74), (OR: 2.43; 95% CI: 1.12–5.26). The other food groups showed no significant association with the absence or presence of hepatic fibrosis (Table 5).

## 4. Discussion

Several research studies have reported an increase in the prevalence of NAFLD with aging [21,22]. However, the mean age of our sample (43.3 ± 0.1 years) was below the average found in other studies [21,22]. In addition, this study highlighted a high prevalence of metabolic syndrome (more than 80%) independently of the NFS score. This value is higher than that determined by Sibai et al., who found that the prevalence of metabolic syndrome among the general population in Lebanon was 31.2% [23]. This feature has been studied by Marchesini et al., who reported hypertriglyceridemia and hyperinsulinemia in patients diagnosed with NAFLD [6]. The latter is usually associated with obesity [24], high waist circumference (cm), and high waist circumference/hip circumference [25]. In this study, about 80% of male NAFLD patients had a waist circumference or waist-to-hip ratio exceeding the WHO recommendations [26]. This reflects an increase in visceral adiposity, which is generally related to a state of insulin resistance responsible for the onset of metabolic syndrome and fatty liver disease [27].

More than eighty-five percent of observed patients have a positive family history of metabolic disorders. Similarly, Chehreh et al. reported a high prevalence of a family history of type 2 diabetes and hypertension in NAFLD patients in their study. This could be of ethnic, environmental, or hereditary origin [28].

As for the environmental characteristics, the rate of sedentary behavior is high in our sample, with only 20% of the sample exercising regularly. It has been shown that low physical activity had been correlated with NAFLD complications [29], while regular exercise has been shown to improve liver enzymes and fatty liver, regardless of the type or frequency of activity [30]. This is also accompanied by an improvement in insulin resistance, blood triglycerides, and liver histology in patients with NASH [30]. The smoking rate was also high in the sample, in concordance with the study made by Suzuki et al., which correlated smoking with increased oxidative stress and liver injury [31]. This smoking rate is higher than the national average of 41.5% that was reported by Sibai et al. following a national survey done in Lebanon in 2010 [32].

Regarding liver fibrosis, no significant differences in clinical and environmental parameters were found between the two NFS score groups, except for the age and the presence of obesity, diabetes type 2, cardiovascular diseases (CVD), and the ratio of ASAT to ALAT. These parameters were found to be associated with the presence of hepatic fibrosis [33], and the ASAT/ALAT ratio could be used as a screening tool for liver evaluation and detection of advanced liver fibrosis [34]. Their increase is related to the presence of free inflammatory fatty acids, which contribute to a state of insulin resistance, hence increasing the degree of severity of non-alcoholic steatosis (Table 3) [35].

Three major dietary patterns were derived from our sample in the current study. The Traditional, the High Fruit pattern, and the Westernized pattern. The presence of the High Fruit pattern among NAFLD Lebanese patients had been discussed in a previous study [19]. This is in accordance with results obtained in other Mediterranean populations, especially that fruits such as apples, plums, and raisins are of low cost and are available in the Lebanese market. Fructose as a main monosaccharide present in fruits had been linked to NAFLD and, more specifically, to fibrosis [36]. An increase in its consumption increases endoplasmic reticulum stress, promotes activation of stress-related kinases, and induces mitochondrial dysfunctions [36].

An interesting result obtained in this study is that NAFLD Lebanese patients were more inclined to consume a traditional diet while aging with an increase in odds by 1.6, *p* < 0.05 (Table 3). This feature is also present in educated people, with an increase in odds of 2.90 towards a traditional diet, versus a decrease by 50% in adherence to a high fruit diet. A study done by Hiza et al. reported that children and older adults had better-quality diets than younger and middle-aged adults with a clearer tendency toward vegetables, whole grains, and legumes [37]. Moreover, adults with a college diploma had a higher adherence to vegetables and whole grains. This indicates an ability to translate nutritional knowledge into better dietary practices [37,38]. Another finding was an increase in BMI following a Westernized diet. Pliego et al. found that the higher the score of unhealthy patterns, the higher the BMI [39]. These results suggest that weight gain is determined by both qualitative and quantitative aspects of food consumption [39].

The multivariate analysis confirmed the hypothesis that a traditional diet was correlated with a low risk of fibrosis. The obtained results showed a significant relationship between high adherence to the traditional diet and the absence of fibrosis with a decreased risk of getting fibrosis by 82%, *p* = 0.031, after adjusting for its covariables (Table 4). Diets enriched in vegetables, legumes, olive oil, seeds, and red wine have proven their beneficial effects on all the risk factors associated with metabolic syndrome and NAFLD [40]. This can be explained through several mechanisms that can vary from an effective dietary approach for weight loss to a model diet that is plentiful in some beneficial nutrients such as antioxidants, vitamins, and monounsaturated fatty acid (MUFA) through the presence of olive oil as the main contributor of fat [41]. A westernized diet characterized by a high intake of pasta, red meat, desserts, and pizza, rich in simple sugar as well as in saturated and trans fatty acids is well known to trigger an increase in weight, higher postprandial insulin secretion and ultimately an increase in liver fat storage [42].

Concerning nutrients, simple carbohydrates were identified as risk factors for NAFLD, increasing the risk of fibrosis twofold, *p* = 0.09 (Table 5). An increase in the intake of simple carbohydrates was also associated with hepatic de novo lipogenesis (DNL) and hepatic inflammation [43,44]. According to Zykovic et al., a reduction in the amount of total carbohydrates, especially simple sugars, would reduce the total pool of acetyl CoA in the liver and, therefore, reduce the flux through the DNL pathway [45]. The reduction in triacylglycerol synthesis would also prevent the excess accumulation of total fat in the liver [46].

The consumption of long-chain polyunsaturated fatty acids was low in the sample. This is realistic, since Lebanese people rarely consume fatty fish. Supplements enriched with EPA, DHA, and ω-3 are rarely used among Lebanese NAFLD patients. Although Lebanon is a coastal country, the Lebanese population avoids eating seafood for cultural, economic, and public health reasons. The general belief is that the coastline is polluted, and consumable fish are unavailable or very expensive [19]. A study done by Nassreddine et al. reported this low consumption of seafood in Lebanese subjects, with 73.6% of Lebanese adult participants consuming less than two servings of fish per week [47]. A medium adherence to EPA and DHA was identified as decreasing fibrosis by approximately 65% and 74%, respectively, *p* < 0.05, as compared to a low adherence after adjustment for the covariables (Table 5). sThese two nutrients are well known for their anti-inflammatory and anti-oxidative properties and their beneficial effect in the treatment of NASH. EPA and DHA converted from ω-3 polyunsaturated fatty acids (PUFAs) might inhibit the accumulation of triglycerides by modifying hepatic lipid metabolism leading to an increase in triglyceride transportation from hepatocytes [46], and could reduce inflammatory and oxidative status [44,46]. In addition, long-chain polyunsaturated fatty acids may induce transcription of genes encoding enzymes for fatty acid oxidation through their ability to act as ligand activators of PPAR-α (peroxisome proliferator-activated receptor-α) [48]. Dietary EPA and DHA were identified as linear, independent, and preventive parameters for NAFLD in Japanese men who generally consume more fish than Western men [49].

Concerning ω-6, a medium intake/day versus a low intake/day was found to increase the risk of fibrosis by 2.24-fold, *p* < 0.05. This is in concordance with other studies which showed an association between ω-6 intake and the development of steatosis and fibrosis in animal models [50,51,52,53,54]. It had been reported that ω-6 metabolism causes oxidative stress and mitochondrial dysfunction. It enhances liver Kupffer cell production of inflammatory cytokines, exacerbates systemic and hepatic insulin resistance, and worsens inflammation and fibrosis [55]. This contributes to the effect of ω-6 intake on the risk of fibrosis as reported in our results after adjustment with the covariables.

Finally, it should be noted that the study had some limitations, such as selection and sampling bias. The sampling bias was mainly due to the difficulty of getting a significant sample of the Lebanese population because the study was done in one outpatient clinic in an academic hospital in Beirut. The second limitation was the ultrasound used to assess the presence or not of fatty liver (sensitivity of ultrasound varies between 60 and 65%, almost no detection for a degree of steatosis <30%) [56]. The third limitation came from recall bias. Patients may overestimate the portion sizes, have memory loss, and over-report their physical activity level. To overcome these biases, patients were interviewed to report their dietary intake prior to the disease diagnosis or prior to any diet change due to medical advice or used medications. Another bias is related to the use of factor analysis, which requires subjective decisions for grouping food, choosing the method of rotation or determining dietary patterns according to their loading factors. However, the results obtained were in line with those obtained in other studies [6,19,57], and highlighted the nutritional profile and dietary patterns of Lebanese NAFLD patients in relation to the absence or presence of hepatic fibrosis.

## 5. Conclusions

Three dietary patterns, the Traditional, the High Fruit, and the Westernized patterns, characterized the nutritional profile of Lebanese NAFLD patients. Fruits were absent from the Traditional dietary pattern and constituted a new High Fruit pattern. The Traditional diet was composed mainly of vegetables, nuts, and legumes, which are high in fibers and antioxidants, fish known to be rich in long-chain polyunsaturated fatty acids, and olive oil, which is high in polyphenols and monounsaturated fatty acids. It was found to be inversely associated with the risk or presence of fibrosis. This could be an effective dietary approach for NAFLD patients. The role of fruits in the progression or prevention of the disease has yet to be determined. Further experimental studies are needed to establish a possible relationship between fruit intake/day and the presence of fibrosis in NAFLD patients. This is important to generate dietary NAFLD guidelines and reach a conclusion on the quantity and type of fruits to be consumed by these patients.

## Figures and Tables

**Figure 1 nutrients-14-02554-f001:**
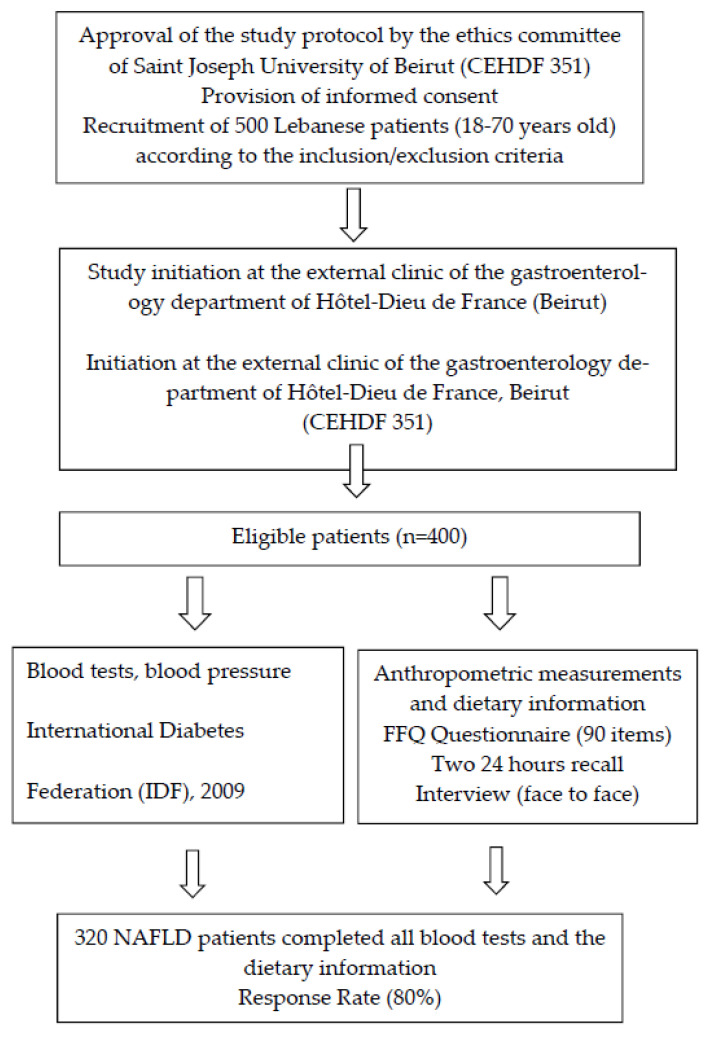
Flow chart for selection and enrolment of the study participants. NAFLD: Non -Alcoholic Fatty Liver Disease, FFQ Food Frequency Questionnaire [11], International Diabetes Federation (IDF) [12].

**Figure 2 nutrients-14-02554-f002:**
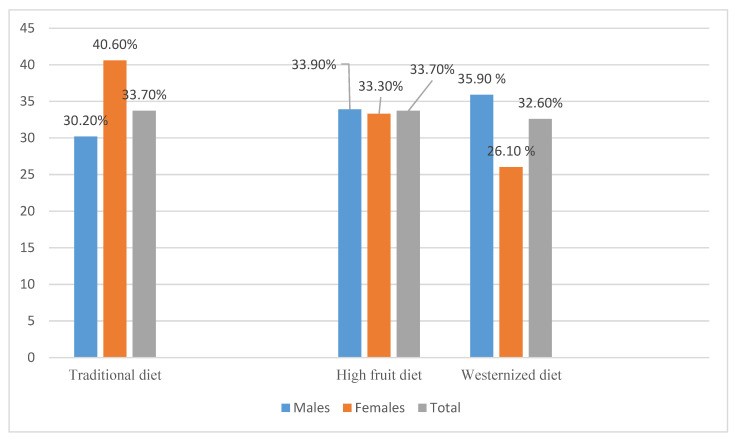
Dietary patterns distribution between males and females (*n* = 320).

**Table 1 nutrients-14-02554-t001:** Sociodemographic, clinical, and environmental parameters according to the NFS scores (*n* = 320).

	NFS Scores		*p*-Value	Total (*n* = 320)
	<−1.45 *n* = 191	≥−1.45 *n* = 129		
Age (years), mean ± SD	40.04 ± 0.11	47.65 ± 0.13	0.0001	43.34 ± 0.12
Sex				
Male	125 (65.5%)	88 (67.9%)	0.778	213 (66.5%)
Female	66 (34.5%)	41 (32.1%)		107 (33.6%)
Place of residence, *n* (%)				
Mount Lebanon	57 (29.8%)	54 (41.9%)	0.314	111 (34.7%)
North	23 (11.9%)	6 (4.7%)		29 (9.1%)
South	14 (7.4%)	9 (6.9%)		23 (7.2%)
Beirut	47 (24.4%)	30 (23.6%)		77 (24.1%)
Bekaa	22 (11.4%)	11 (8.5%)		33 (10.3%)
Nabatieh	17 (9.1%)	13 (10.1%)		30 (9.3%)
Abroad	11 (5.7%)	6 (4.7%)		17 (5.3%)
Marital Status, *n* (%)				
Single	42 (22.0%)	24 (18.9%)	0.853	66 (20.6%)
Married	144 (75.4%)	100 (77.5%)		244 (75.9%)
Divorced	3 (1.6%)	4 (2.8%)		7 (2.10%)
Widow/er	2 (1.04%)	1 (0.8%)		3 (1.10%)
Academic level, *n* (%)				
Illiterate	1 (0.6%)	2 (1.5%)	0.329	3 (0.9%)
Elementary	15 (8.0%)	18 (13.9%)		33 (10.3%)
Intermediate, Secondary	72 (37.4%)	43 (33.3%)		115 (35.9%)
University	103 (54.0%)	66 (51.2%)		169 (52.8%)
Occupation, *n* (%)				
Self-employed	85 (44.6%)	53 (41%)	0.001	138 (43.1%)
Employee	60 (31.4%)	40 (31.4%)		100 (31.2%)
Retired/unemployment	36 (18.9%)	32 (24.7%)		68 (21.2%)
Others	10 (5.2%)	4 (3.1%)		14 (4.4%)
Crowding index ^†^, *n* (%)				
≤1	144 (75.4%)	103 (79.8%)	0.53	247 (77.1%)
>1	47 (24.6%)	26 (20.2%)		73 (22.9%)
Presence of metabolic syndrome, *n* (%)	155 (81.2%)	114 (88.5%)	0.156	269 (83.9%)
Obesity (yes), *n* (%)	103 (53.9%)	101 (78.2%)	0.001	204 (63.7%)
CVD (yes), *n* (%)	8 (4.2%)	21(16%)	0.0001	29 (9.1%)
Family medical history (yes), *n* (%)	167 (87.6%)	104 (81.0%)	0.182	271 (84.7%)
Smoking (yes), *n* (%)	78 (40.8%)	55(42.6%)	0.187	133 (41.5%)
Physical Activities (Kcals)(yes)	41 (21.46%)	23 (17.82%)	0.469	64 (20%)
Energy intake (kcal), (M), mean ± SD	4525.8 ± 0.2	4127.6 ± 0.2	0.110	4162.9 ± 0.2
Energy intake (kcal), (F), mean ± SD	2829.4 ± 0.2	2731.5 ± 0.3	0.737	2747.3 ± 0.3
Waist circumference (cm) (M)≥ 94 ^§^ (F) ≥80 ^§^	130 (68.2%) 180 (94.5%)	107 (83.0%) 127 (98.7%)	0.006 0.129	237 (74.2%) 307 (96.0%)
Waist/hip ratio (M) > 0.90 ^§^ (F) > 0.85 ^§^	144 (75.6%) 171 (89.8%)	103 (79.6%) 124 (96.1%)	0.438 0.058	247 (77.3%) 295 (92.2%)
Diabetes type 2, *n* (%)	25 (13.0%)	43 (33.0%)	0.0001	68 (21.2%)
Homa > 3, *n* (%)	92 (48.3%)	66 (51.5%)	0.697	158 (49.4%)
ASAT/ALAT ≥ 1	29 (15.3%)	34 (26.4%)	0.032	63 (19.7%)
Current dietary ** supplementation use, *n* (%)	81 (42.4%)	70 (54.3%)	0.053	151 (47.2%)

Continuous variables were reported as geometric means± standard deviations. Categorical variables were reported as numbers and percentage. Statistical tests used: independent, *t*-test (continuous variables), test χ2-test (categorical variables). SD, standard deviation, *p* < 0.05. ^†^ crowding index: amount of residents/number of rooms. ^§^ waist circumference and waist to hip ration: values according to the IDF, 2009 (M/F). ** Current dietary supplementation use; 75% of supplements were Vit D.

**Table 2 nutrients-14-02554-t002:** Factor loading matrix for the three identified dietary patterns in the study population.

Food Group	Pattern		
	Traditional Lebanese	High Fruits	Westernized
Vegetables	0.85		
Chickpeas, red beans, lentils, peas	0.50		
Vegetable oil/olives	0.33		0.20
Almonds, walnuts, hazelnuts, sesames	0.27		
Fish	0.21		
Sea Food			0.36
Red Wine	0.21		
Fruits and fruit juices		0.73	
Hamburger and fries			0.63
Beef meat			0.53
Chicken			0.53
Carbonated beverages			0.52
Pizza			0.52
Chips			0.50
Pork			0.41
Hot Dog			0.45
Ketchup			0.50
Mayonnaise or mustard			0.41
1 chicken egg			0.44
Spaghetti or noodles			0.41
Cooked rice	0.24		0.39
Pies or fatayer			0.39
Bread			0.27
Desserts, Arabic pastries			0.30
Milk chocolate			0.39
Laban/Lebanese yogurt			0.25
Energy drink			0.22
Beer			0.21
Pop Corn			0.20
Percent variance explained by each pattern	18.22%	9.84%	7.22%

Extraction method: principal component analysis; Rotation method: Varimax with Kaiser normalization; Absolute values ≤ 0.2 were excluded from the table.

**Table 3 nutrients-14-02554-t003:** Association between dietary patterns, sociodemographic and clinical characteristics in the study population.

	Traditional Lebanese		High Fruits		Westernized	
	OR	CI	OR	CI	OR	CI
Age (years)	1.60	1.03–1.09 *	1.01	0.98–1.03	0.92	0.89–0.95 *
BMI (kg/m^2^)	0.90	0.91–1.02	0.95	0.89–1.01	1.11	1.04–1.18 *
Education ** (university level/others)	2.90	1.50–5.76 *	0.50	0.27–0.91 *	0.76	0.39–1.48

* Test statistic; Multivariate -adjusted OR (95% CI) using binary logistic regression, *p* < 0.05. The Model was adjusted for gender, crowding index, presence of metabolic syndrome (no/yes), physical activity (yes/no), family history (no/yes), marital status (married, single, widow), smoking (no/yes) and profession (freelance, employee, unemployed, retirement and others). ** Education (university level versus illiterate, primary, secondary, and high school level).

**Table 4 nutrients-14-02554-t004:** Hepatic fibrosis across tertiles (T) of dietary pattern scores.

Dietary Pattern	T1 (Low Adherence)	T2 (Medium Adherence)	T3 (High Adherence)	*p*-Trend
Traditional Lebanese				
Crude Model	Ref *	0.41 (0.15–1.12)	0.36 (0.13–1.03)	0.057
Model 1	Ref	0.31 (0.99–1.18)	0.21 (0.07–0.82) ^‡^	0.024
Model 2	Ref	0.32 (0.07–1.21)	0.18 (0.04–0.79) ^‡^	0.023
Model 3	Ref	0.42 (0.11–1.96)	0.18 (0.04–0.85) ^‡^	0.031
High Fruits				
Crude Model	Ref	1.22 (0.42–3.53)	0.98 (0.32–3.02)	0.969
Model 1	Ref	1.68 (0.39–7.11)	1.05 (0.24–4.69)	0.943
Model 2	Ref	1.59 (0.32–7.93)	1.06 (0.21–5.35)	0.946
Model 3	Ref	2.53 (0.43–14.97)	1.78 (0.28–11.17)	0.537
Westernized				
Crude Model	Ref	1.51 (0.47–4.82)	1.03 (0.31–3.39)	0.956
Model 1	Ref	1.78 (0.53–6.01)	1.41 (0.38–5.28)	0.606
Model 2	Ref	2.43 (0.59–9.99)	1.19 (0.28–5.02)	0.813
Model 3	Ref	2.22 (0.51–9.61)	1.04 (0.24–4.51)	0.959

^‡^ Test statistic; Multivariate -adjusted OR (95%CI) using binary logistic regression, *p* < 0.05. The Model 1 was adjusted for age, gender, and the crowding index. Model 2; Model 1 + presence of metabolic syndrome (no/yes), physical activity (yes/no), obesity (no, yes), diabetes type 2 (no, yes), family history (no/yes) and smoking (no/yes). Model 3; Model 2 adjusted for marital status (married, single, widow), education (illiterate, primary, high school, and university) and profession (freelance, employee, unemployed, retirement and others). * Ref referred to the first tertile of dietary pattern (low adherence to the corresponding dietary pattern).

**Table 5 nutrients-14-02554-t005:** Hepatic fibrosis across tertiles of food groups.

Food Groups	Tertile 1	Tertile 2	Tertile 3	*p*-Trend
Fructose (g/day)	Ref *	1.07(0.52–2.20)	1.15(0.54–2.45)	0.718
Fibres (g/day)	Ref	0.67(0.32–1.39)	1.24(0.61–2.52)	0.554
Monounsaturated Fatty acids (g/day)	Ref	0.91(0.45–1.84)	0.69(0.33–1.46)	0.342
Polyunsaturated Fatty acids (g/day)	Ref	0.77 (0.39–1.52)	0.65(0.31–1.36)	0.253
Saturated Fatty acids (g/day)	Ref	1.50 (0.74–3.06)	0.83(0.38–1.83)	0.653
ω-3 (g/day)	Ref	1.12 (0.55–2.29)	1.03(0.49–2.16)	0.944
ω-6 (g/day)	(1.20–13.70) Ref	(13.71–19.01) 2.24 (1.06–4.74) ^§^	(19.02–52.70) 1.04(0.50–2.14)	0.922
Cholesterol (mg/day)	Ref	0.79 (0.39–1.61)	1.10(0.52–2.30)	0.792
Protein (g/day)	Ref	0.91 (0.43–1.91)	0.65(0.28–1.50)	0.311
Fat (g/day)	Ref	1.06 (0.50–2.23)	0.55(0.24–1.26)	0.159
Simple Carbohydrates (g/day)	(8.15–101.4) Ref	(101.50–164.69) 2.43 (1.12–5.26) ^§^	(164.70–469.64) 2.06(0.87–4.88)	0.099
EPA (mg/day)	(0.1–2.0) Ref	(2.10–5.0) 0.35 (0.15–0.82) ^§^	(5.1–160) 0.65(0.26–1.63)	0.357
DHA (mg/day)	(0.1–1.0) Ref	(1.1–9.0) 0.26 (0.09–0.73) ^§^	(9.1–230) 0.53 (0.19–1.44)	0.212
Energy/day (Kcals), males	Ref	0.63 (0.26–1.57)	1.01 (0.38–2.64)	0.994
Energy/day (Kcals), females	Ref	2.27 (0.48–10.80)	1.53 (0.33–7.09)	0.587

Test statistic; ^§^ Multivariate -adjusted OR (95%CI) using binary logistic regression, *p* < 0.05. The Model is adjusted for age, gender, crowding index, presence of metabolic syndrome(no/yes), obesity (no, yes), diabetes type 2 (no, yes) physical activity (yes/no), family history (no/yes), marital status (married, single, widow), smoking (no/yes), education (illiterate, primary, high school, and university) and profession (freelance, employee, unemployed, retirement and others). * Ref referred to the first tertile of nutrients consumed (g/day).

## Data Availability

Details on FFQ questionnaire, food groups can be found in Appendix A. Data sets can be provided upon requests.

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
