# Peer review of "Relation of Dietary Patterns and Nutritional Profile to Hepatic Fibrosis in a Sample of Lebanese Non-Alcoholic Fatty Liver Disease Patients"

_nutrients, 2022, doi:10.3390/nu14122554_

Round 1

Reviewer 1 Report

This article is interesting and well organized. The experimental procedures are well explained and the reasoning that supports the experimental design is also properly explained.

Below, you will find some comments and questions that aim to contribute to improving the way information is presented and discussed.

Although referred in the MS, the appendices are not included in the pdf file received so I did nor have access to their content.

Lines 52-54: The formula should not be presented as part of the text, but separately. The units of each variable can also be presented separately to simplify the formula.

Line 59: “dietary pattern” should be “dietary patterns”

Figure 1: Although this figure presents the “course of the study”, it is surprising to see that the approval of the study protocol happened after 500 patients were recruited, after providing their consent. Is there any reason for this? Also, when presenting the “blood tests, blood pressure”, why do the authors include a reference (which is not in the reference list)? Maybe, if it is necessary, it should be included/explained in the figure legend. Additionally, no references are included in relation to the FFQ Questionnaire, for example.

Line 126: correct letter format/size

Line 130: “Blood” does not need capital letter

“2.4 Measuring tools”: authors refer that a food frequency questionnaire was translated into Arabic. Even though it is mentioned that the validity and reproducibility was testes in a previous work, I would like to know if the translation was validated previously in some way? If so, it should be mentioned, and the validation procedure should be presented/referred.

Lines 175-176: “according to WHO 175 obesity classification, 2004”, please include a reference in the MS and in the reference list.

Line 182: “International Diabetes Federation (IDF 2009)”, please include a reference in the MS and in the reference list (the same that is mentioned in figure 1?).

Results: At the beginning of the results section, it would be worth recalling the meaning of "NFS", including it in the title of section 3.1.

 Lines 225: “… (Table 1), the risk of fibrosis was significant according to age...”. I suggest “… (Table 1), and the risk of fibrosis was significant according to age...”

Table 1: correct letter format/size and line spacing in the table footnotes.

3.2. Dietary Patterns: I suggest that this section begins with a more detailed explanation of each of the three types of diet, presenting all their characteristics, as they were considered for the division of study participants. The alternative could be to present these concepts in a subsection of the methods. Table 2 presents this information in another format but, not being explained in more detail, it may not be informative enough for all potential readers of this article.

Just mentioning that the "Traditional diet" consisted of "vegetables, nuts, vegetables, and olive oil" and then not mentioning what is considered as "High Fruit" diet or "Westernized" diet is little information to consider in the analysis of results.

Section 3.3: mentioning the statistical treatment in the title of this section is unnecessary and makes the title more confusing I suggest “Association between dietary patterns and sociodemographic, environmental, and clinical characteristics in the study population” leaving the odds ratio and confidence intervals to be presented/explained in the text. The same can be applied to the following sections.

Tables 4 and 5: What does “Ref” stands for?

Line 303: “There are several research that reported…”. Do you mean “There are several research studies that reported…”?

When referring to the name of authors of other studies in the text, they should only indicate the surname (followed by et al., if applicable) and not put the initial of the first name. For examples, see lines 351 and 356. Please revise the whole MS to check similar situations.

Lines 353-354: How is it that having a university degree is synonymous with greater nutritional knowledge that can be translated into better dietary practices? Does every type of academic degree confer knowledge of nutrition?

Lines 378-381: “Supplements enriched with EPA, DHA and ω-3 are rarely used among Lebanese NAFLD patients.” Is this statement related to the results obtained in the present study? Considering the way this is presented, it is not clear if these are results obtained by the authors or it is based on reference 43. Additionally, can we really conclude that this “indicates” a high consumption of red meat? Eating less fish and seafood does not always mean that people consume red meat. Please revise.

The authors present some reasons that may explain some of the results presented. For example, they mention the possible effects of consuming fatty acids or high amounts of fructose (fruit). Although some reference is made to the beneficial effects of the main components of the traditional diet (lines 346-368), some further discussion on the benefits of consuming the traditional diet and/or the negative effects of consuming the Westernized diet could be interesting, focusing on the mechanisms of action of the main nutrients present in these diets.

The authors finish the discussion by presenting the limitations of the presented study but do not present any strengths. There are none?

The conclusion is mainly based in the results related to fruit consumption. What about other nutrients that may be found in the diets studied?

Additionally, in the conclusions, the authors use the term “pulses” (line 420) which is a synonym of legumes. I suggest that the author use only one designation throughout the whole MS for consistency.

Regarding the cited references, more than 2/3 of the references were published more than 10 years ago and only 5 references (less than 10%) were published in the last 5 years. Does this mean that this topic is no longer interesting to researchers or do the references need to be updated? Moreover, I suggest that you revise the presented list because references are numbered twice.

Reviewer 2 Report

 I have some minor comments:

1)    The last two sentences in the Abstract should put in the Discussion, and the significance of this research should be added.

2)    The sentence “ Multivariate analysis showed a significant relationship between high/ low adherence to the traditional diet and absence of hepatic fibrosis with an odd of 0.18” was not clear.

3)    How to correlate food groups in table 5 and three dietary patterns?

4)    The Introduction mentioned 3 kind of NFS score(<-1.45; -1.45-0.67; and >0.67), why the authors did not apply the same standard to group their samples?

5)    The font and style of the text in the whole manuscript should be the same.

6)    The reference style was not in line with the journal.
